# Network of Interactions between ZIKA Virus Non-Structural Proteins and Human Host Proteins

**DOI:** 10.3390/cells9010153

**Published:** 2020-01-08

**Authors:** Volha A. Golubeva, Thales C. Nepomuceno, Giuliana de Gregoriis, Rafael D. Mesquita, Xueli Li, Sweta Dash, Patrícia P. Garcez, Guilherme Suarez-Kurtz, Victoria Izumi, John Koomen, Marcelo A. Carvalho, Alvaro N. A. Monteiro

**Affiliations:** 1Cancer Epidemiology Program, H. Lee Moffitt Cancer Center and Research Institute, Tampa, FL 33612, USA; golubeva.3@osu.edu (V.A.G.); thales.cn@gmail.com (T.C.N.); xueli.li@moffitt.org (X.L.);; 2Divisão de Pesquisa Clínica, Instituto Nacional de Câncer, Rio de Janeiro 20230-130, Brazil; gregoriis@gmail.com (G.d.G.); Kurtz@inca.gov.br (G.S.-K.); 3Departamento de Bioquímica, Instituto de Química, Federal University of Rio de Janeiro, Rio de Janeiro 21941-909, Brazil; rdmesquita@iq.ufrj.br; 4Cancer Biology PhD Program, University of South Florida, Tampa, FL 33612, USA; 5Institute of Biomedical Science, Federal University of Rio de Janeiro, Rio de Janeiro 20230-130, Brazil; ppgarcez@gmail.com; 6Proteomics and Metabolomics Core, H. Lee Moffitt Cancer Center and Research Institute, Tampa, FL 33612, USA; victoria.izumi@moffitt.org; 7Chemical Biology and Molecular Medicine Program, H. Lee Moffitt Cancer Center and Research Institute, Tampa, FL 33612, USA; john.koomen@moffitt.org; 8Instituto Federal do Rio de Janeiro-IFRJ, Rio de Janeiro 20270-021, Brazil

**Keywords:** ZIKV, protein–protein interaction, non-structural viral proteins, network

## Abstract

The Zika virus (ZIKV) is a mosquito-borne Flavivirus and can be transmitted through an infected mosquito bite or through human-to-human interaction by sexual activity, blood transfusion, breastfeeding, or perinatal exposure. After the 2015–2016 outbreak in Brazil, a strong link between ZIKV infection and microcephaly emerged. ZIKV specifically targets human neural progenitor cells, suggesting that proteins encoded by ZIKV bind and inactivate host cell proteins, leading to microcephaly. Here, we present a systematic annotation of interactions between human proteins and the seven non-structural ZIKV proteins corresponding to a Brazilian isolate. The interaction network was generated by combining tandem-affinity purification followed by mass spectrometry with yeast two-hybrid screens. We identified 150 human proteins, involved in distinct biological processes, as interactors to ZIKV non-structural proteins. Our interacting network is composed of proteins that have been previously associated with microcephaly in human genetic disorders and/or animal models. Further, we show that the protein inhibitor of activated STAT1 (PIAS1) interacts with NS5 and modulates its stability. This study builds on previously published interacting networks of ZIKV and genes related to autosomal recessive primary microcephaly to generate a catalog of human cellular targets of ZIKV proteins implicated in processes related to microcephaly in humans. Collectively, these data can be used as a resource for future characterization of ZIKV infection biology and help create a basis for the discovery of drugs that may disrupt the interaction and reduce the health damage to the fetus.

## 1. Introduction

Zika virus (ZIKV) is a neurotropic arthropod-borne virus belonging to Flaviviridae family, along with other Flaviviruses capable of infecting central nervous system, such as West Nile Virus, St. Louis Encephalitis Virus, and Japanese Encephalitis Virus. It is commonly transmitted though the bite of an infected *Aedes aegypti* mosquito. Importantly, besides the mosquito bites, human-to-human modes of transmission have also been documented, including sexual activity, blood transfusions, and mother to fetus [1].

Since its first confirmed human infection in the 1960s, there were three documented Zika virus (ZIKV) outbreaks worldwide. The first two occurred in Micronesia and French Polynesia in 2007 and 2013, respectively. The most recent one (2015–2016) started in the northeastern region of Brazil and rapidly spread through South America, the Caribbean, and Mexico. By July 2016, locally transmitted cases of Zika infection were first reported in the United States (Florida). According to the World Health Organization (WHO), 73 different countries had reported ZIKV infections by February of 2016 [2,3]. According to the Centers for Disease Control & Prevention, there have been no recorded local transmissions of the Zika virus in the continental United States in 2018 and 2019. However, with the globally increasing rate of travelling and the historical ability of viruses to acquire genetically modified virulence, the search for effective methods of Zika prevention and control remains important.

ZIKV infections in adults have been associated with neurological conditions such as Guillain-Barré syndrome, acute flaccid paralysis, and meningoencephalitis [4,5,6,7]. The Brazilian outbreak was the first time that ZIKV infection (presented in pregnant women) was correlated to congenital microcephaly in newborns [8,9]. Both in vitro and in vivo models have demonstrated that ZIKV has a tropism toward human neural progenitor cells [10,11,12]. In these cells, ZIKV infection is followed by apoptosis, corroborating the hypothesis of ZIKV as the etiological agent of these neurological disorders [4,5,10,11,12]. Further, independent studies have shown that the microcephaly and neural development-associated phenotypes is not a distinct feature of the Asian lineage [12,13,14,15,16]. However, the precise molecular mechanism(s) underlying these ZIKV-related manifestations is not understood.

ZIKV is a Baltimore class IV arbovirus from the Flaviviridae family. The ZIKV genome encodes a polyprotein that is processed by both viral and host proteases into ten proteins. Three of them (the capsid, pre-membrane, and envelope) are responsible for the structural organization of the virus. The other seven are non-structural (NS) proteins (NS1, NS2A NS2B, NS3, NS4A, NS4B, and NS5) responsible for regulatory function, viral replication, and subvert host responses [17].

The identification of virus–host protein–protein interaction is essential to better understand viral pathogenesis and to identify cellular mechanisms that could be pharmacologically targeted [18]. To gain further insight into the ZIKV pathogenesis, we generated a virus–host protein–protein interaction network focused on the interactions mediated by the non-structural proteins encoded by the Brazilian ZIKV genotype. Here, we present a network composed of proteins related to neuron projection development, microcephaly-associated disorders, and by protein complexes linked to replication and infection of other members of the Flaviviridae family. In addition, we integrate our dataset with previously published ZIKV protein interaction networks, highlighting common and unique protein interaction partners [19,20,21]. In addition, we show a PIAS1-dependent control of NS5 protein stability. Taken together, these data can be used as a resource to improve the understanding of the ZIKV pathogenesis and identify putative pharmacological targets for future treatment approaches.

## 2. Materials and Methods

### 2.1. cDNA Constructs

We generated the cDNA of seven individual NS proteins (NS1, NS2A, NS3, NS2B, NS4A, NS4B, and NS5) corresponding to a ZIKV Brazilian isolate from the state of Pernambuco (GenBank AMD16557.1) [22]. We used the virus isolate as template for polymerase chain reaction (PCR) to generate cDNAs for NS1, NS2A, NS2B, NS4A, and NS4B. We were unable to obtain the correct product corresponding to NS3 and NS5 cDNAs using the virus isolate. We generated the Brazilian genotype cDNAs for NS3 and NS5 using nucleotide substitutions introduced by site-directed mutagenesis on the Asian lineage (PRVABC59 strain) cDNAs for NS3 and NS5 (pLV_Zika_Flag_NS3 and pLV_Zika_NS5_Flag plasmids; Addgene #79634 and #79639, respectively).

ZIKV cDNAs coding for NS proteins were cloned into pGBKT7 (Clontech) in frame with the GAL4 DNA binding domain (DBD) for yeast-two hybrid (Y2H) assays, and into pNTAP (Agilent) in frame with the Streptavidin-binding peptide (SBP) and Calmodulin-binding peptide (CBP) epitope tags for the tandem affinity purification coupled to mass spectrometry (TAP-MS) assays. The glutathione-S-transferase (GST)-tagged baits used for Y2H validations were generated by subcloning the cDNA from the isolated pGADT7 (Clontech) plasmid into the pDEST27 vector using Gateway recombination cloning according to the manufacturer’s instructions (ThermoFisher).

To validate Y2H interactions, recovered Y2H plasmids containing prey cDNAs were amplified by PCR using primers containing attb sites. PCR products were cloned into pDONR221 for Gateway recombination cloning (Invitrogen) and subsequently into pDEST27, to produce an N-terminal GST fusion.

PIAS1 cDNA (NM 001320687.1) was obtained via PCR amplification using a human leukocyte cDNA library and was cloned into pGEX-6P1 using *EcoRI* and *SalI* sites. For expression in mammalian cells, PIAS1 cDNA was subcloned into the pEBG vector using *Bam*HI and *NotI* sites. All constructs were confirmed by Sanger sequencing.

### 2.2. Y2H Library Screening

To identify direct human brain protein targets of ZIKV NS proteins, we used the MATCHMAKER Gold Y2H system (Clontech). Seven ZIKV viral proteins (NS1, NS2A, NS2B, NS3, NS4A, NS4B, and NS5) were transformed into the Saccharomyces cerevisiae strain Y2HGold (Clontech) alone or co-transformed together with an empty pGADT7 vector and tested for auto-activation and toxicity (defined by low transformation efficiency, small colony phenotype, or inability to grow in liquid culture), as previously described by our group [23].

All bait proteins were expressed in Y2HGold and did not induce toxic effects on the yeast cell cycle or survival (Appendix A). Y2HGold transformants expressing each bait were mated to Y187 strain expressing a pre-transformed human brain normalized cDNA library (Matchmaker^®^ Gold Yeast Two-Hybrid System; Cat.no. 630486; Clontech) for 20 h. The mated cultures were then plated on quadruple dropout medium (SD -Trp/-Leu/-His/-Ade) and incubated for 8–12 days (NS5 was screened twice). For every screen, more than 1 × 10^6^ transformants were screened (Appendix A). Yeast miniprep DNA was used to recover pGADT7 fusions from each positive clone (Clontech Yeast Plasmid Isolation Kit), amplified by KOD polymerase chain reaction (PCR) and Sanger sequenced using a T7 primer. Out of frame clones were discarded and in-frame clones were kept for further analysis (Appendix A).

### 2.3. Validation of Y2H Interactions

Protein–protein interactions identified in Y2H screens were validated by expression in human embryonic kidney (HEK) 293FT cells and protein pulldowns. HEK293FT cells were co-transfected with pDEST27 containing prey fusions to GST, and pNTAP containing bait fusions to SBP and CBP. Cells were collected after 24 h and lysed in 1% 3-[(3-Cholamidopropyl)dimethylammonio]-1-Propanesulfonate (CHAPS) lysis buffer (1% CHAPS, 150 mM NaCl, 10 mM 4-(2-hydroxyethyl)-1-piperazineethanesulfonic acid (HEPES) (pH 7.4) with protease and phosphatase inhibitors). Whole cell lysates were subjected to affinity purification of the TAP-tagged NS constructs using streptavidin-conjugated agarose beads, which were washed four times with 1% CHAPS lysis buffer, and then analyzed by Western blot using anti-GST (GE27–4577-01; Sigma Aldrich) and anti-CBP tag antibodies (GenScript; Cat.no. A00635).

### 2.4. Tandem Affinity Purification Coupled to Mass Spectrometry

HEK293FT cells were transfected using the calcium phosphate method with the SBP-CBP-tagged NS or control (Green Fluorescent Protein; GFP) vectors (Figure 1A). HEK cells have been previously used as a model for characterizing host–pathogen protein–protein interactions (PPIs) [20,24,25].

About 1 × 10^8^ cells were used for the purification of the protein complexes using the InterPlay TAP purification kit (Stratagene) as described previously [23].

A nanoflow ultra-high-performance liquid chromatograph (RSLC, Dionex) coupled to an electrospray bench top orbitrap mass spectrometer (Q-Exactive plus, Thermo) was used for liquid chromatography-tandem mass spectrometry (LC-MS/MS) peptide sequencing experiments. Samples were loaded onto a pre-column and washed for 8 min with aqueous 2% acetonitrile and 0.04% trifluoroacetic acid. Trapped peptides were eluted onto an analytical column (C18 PepMap100, Thermo) and separated using a 90-min gradient delivered at 300 nl/min. Sixteen tandem mass spectra were collected in a data-dependent manner following each survey scan using a 15 s exclusion for previously sampled peptide peaks (QExactive, Thermo).

### 2.5. Analysis of Proteomics Data

We used Scaffold (Version 4.8.5) to obtain the original samples report of all TAP-MS based peptide and protein identifications. Peptide identifications were retained if they satisfied a minimum of 95.0% threshold. Protein identifications were accepted if they met greater than 50.0% threshold with a minimum of two identified peptides.

To further analyze the original Scaffold mass spectrometry data, we used APOSTL, an integrative Galaxy pipeline for affinity proteomics data [26]. The following global cutoffs were applied to 7996 interactions and generated a list of 88 high confidence interactions: SaintScore cutoff: 0.5; FoldChange cutoff: 0; normalized spectral abundance factor (NSAF) score cutoff: 0.0000025.

APOSTL also interrogates the CRAPome database (http://crapome.org/), which contains common contaminants in affinity purification–mass spectrometry data [27]. Seventeen proteins displayed a CRAPome score >90% and were candidates to be called non-specific interactors. However, two hits were plausible as they were previously implicated in microcephaly (RAB18 and NEDD1), and two hits were found to be associated with ZIKV NS proteins in a previously published independent study (AHCYL1 and GET4) [20]. Moreover, only 2 out of 17 displayed multiple interactions, suggesting that the other 15 proteins do not constitute non-specific interactors in our assay. Therefore, we decided to retain all proteins, but have indicated the high CRAPome score in Figure 2 when appropriate.

### 2.6. Generation of the Microcephaly-Associated Protein–Protein Interaction Network (PIN)

We generated a microcephaly-associated PIN by searching National Center for Biotechnology Information (NCBI) ENTREZ Gene using [microcephaly] AND [Homo sapiens] as a query. This exercise led to 277 genes, which were manually curated to remove those without an Online Mendelian Inheritance in Man (OMIM) designation (i.e., pseudogenes and partially characterized loci) with a final tally of 261 genes. These genes were used as input to BisoGenet [28], which adds edges between the input nodes, to generate a microcephaly-associated network with 370 interactions (Appendix A). BisoGenet settings were ‘input nodes only’ (Methods) and checking ‘protein–protein interactions’ only leaving ‘Protein DNA interaction’ and ‘microRNA silencing interaction’ unchecked. Significant interaction clusters were identified using ClusterONE (Version 1.0) [29] using the following settings: ‘minimum size’ = 5; ‘minimum density’ = 0.5; ‘edge weights’ = unweighted. Gene ontology (GO) enrichment of clusters was done using BINGO [30] as a Cytoscape plug-in.

### 2.7. Network Generation and GO Analysis

Network graphics were generated with Cytoscape version 3.7.1 [31]. Each NS integrated dataset was analyzed using WebGestalt to determine the enrichment of GO terms. For each bait set (all proteins that interact with each NS bait), the number of genes in the set that was scored for a term was obtained. The number was then divided by the number of genes in the GO database for that term to obtain an enrichment ratio. Enrichment ratios were log2-transformed to depict increase and decrease changes as numerically equal, but with an opposite sign. To allow for log transformation, enrichment ratios with a 0.0 value were replaced by half of the lowest non-zero value in the complete set. Bait sets were clustered with Cluster 3.0 using the Correlation (uncentered) metric of similarity with no filtering, and the clustering method chosen was complete linkage. It was visualized using Java TreeView v 1.1.6r4 [32].

### 2.8. Mitocheck Analysis and Clustering

The Mitocheck phenotype database (20,921 genes), which scores 14 mitosis-related phenotypes in a binary form (presence = 1; absence = 0) from RNA interference screens, was downloaded from http://www.mitocheck.org/ as a tab-delimited file, in which genes are represented in rows and phenotypes in columns.

The enrichment and clustering (for each bait set) were performed as described above. Further, we also deconvoluted the integrated dataset to genes that were positive to at least one of the 14 phenotypes (Appendix A). These new data sets were clustered with Cluster 3.0 using the Correlation (uncentered) metric of similarity with no filtering, but log2-transformed to depict increase and decrease changes as numerically equal, but with an opposite sign. The clustering method chosen was complete linkage. It was visualized using Java TreeView v 1.1.6r4 [32].

### 2.9. GST Pulldown Assay

HEK293FT cells were transfected using Polyethylemine (PEI; Polysciences Inc.) as previously described [33]. GST pulldown assays were performed by incubating for 16h at 4 °C Glutathione Sepharose 4B (GE Healthcare) with whole cell lysates prepared 48 h after transfection. The resin was extensively washed with ‘mild’ Radioimmunoprecipitation Assay (RIPA) buffer (50 mM Tris-Cl pH 7.4, 150 mM NaCl, 1 mM Ethylenediaminetetraacetic acid, 1% NP40, and 2.5 mM Dithiothreitol), boiled in loading buffer, and analyzed by Western blotting using anti-GST (Santa Cruz Biotechnology; B-4) and anti-GFP (Millipore; MAB3580) antibodies.

### 2.10. Protein Stability Assay

HEK293FT cells were transfected with expression vectors containing GST-tagged PIAS1 and GFP- or CBP-tagged NS5 (empty expression vectors were used as negative controls) and treated 24 h later with 10 µg/mL of cyclohexamide (or Dimethylsulfoxide) for varying lengths of time. Whole cell lysates were analyzed by Western blotting using anti-GST, anti-CBP, or anti-GFP and anti-β-actin (Santa Cruz Biotechnology; C4).

## 3. Results

### 3.1. Yeast-Two Hybrid Screenings

To build the first pair-wise protein–protein interaction database of ZIKV NS proteins encoded by the Brazilian genotype, we performed stringent yeast-two hybrid (Y2H) screenings using ZIKV NS proteins as baits to screen a normalized human brain cDNA library (Figure 1A and Appendix A). The Y2H screenings generated a protein–protein interaction network (PIN) ranging from 1 (NS2A) to 56 (NS3) interactions, totaling 99 unique protein hits and 109 bait–prey interactions (Figure 1A,B and Appendix A).

To validate our Y2H PIN, a subset of 38 bait–prey interactions (38.4% of the Y2H PIN) was tested for interaction in human HEK293FT cells. All NS coding sequences were cloned in a eukaryotic expression vector in frame with the streptavidin and calmodulin binding peptides (SBP and CBP, respectively). Candidate interactor cDNAs were expressed in frame with glutathione-S-transferase (GST). SBP pulldown assays were performed against GST-tagged preys in HEK293FT cells, and 76.3% of interactions were confirmed (Figure 1C). As some true interactors might not validate in these specific conditions, Figure 1C retains all interactions, with validated ones indicated. However, further analysis was conducted, retaining only the validated hits.

### 3.2. Tandem Affinity Purification Followed by Mass Spectrometry

To further characterize the NS-mediated protein interactions, we expressed all baits as fusions to SBP and CBP in HEK293FT cells (Figure 1A). We then performed tandem affinity purification coupled to mass spectrometry (TAP-MS), which resulted in a high-confidence PIN with interactions ranging from 8 (NS2B and NS5) to 27 (NS2A), totaling 62 unique protein hits and 89 bait–prey interactions (Figure 1A and Figure 2A,B; Appendix A).

### 3.3. Merged ZIKV PIN

We combined the final Y2H and TAP-MS networks to generate the Merged ZIKV Network containing 157 nodes (including the baits) with 189 interactions between NS proteins and human host proteins (Figure 3A) (Appendix A). Only 3 out of 151 hits were common to both Y2H and MS-based networks (BCLAF1, AHCYL1, and COPB1), indicating a limited overlap between methods.

### 3.4. Gene Ontology

Gene ontology (GO) enrichment analysis of the Merged ZIKV PIN identified a subset of proteins mainly involved in 13 non-redundant biological processes (Appendix A). Among the hits identified, 13 are members of the proteasome complex (11 unique to the 26S subunit) and five members of the chaperonin-containing TCP1 (CCT) complexes (8.7% and 3.3% of our PIN, respectively) (Appendix A and Appendix A). GO enrichment analysis of cellular components revealed an enrichment of peptidase complex, chaperone complex, and ficolin-1-rich granules (Appendix A).

Next, we applied unsupervised clustering of bait sets according to their GO enrichment ratios for biological processes and cellular components (Figure 3B,C and Appendix A). Interestingly, protein bait sets were clustered into two major groups (NS1, NS2A and NS3 versus NS2B, NS4A, NS4B, and NS5).

### 3.5. Phenoclusters

Autosomal recessive primary microcephaly (MCPH) development is intrinsically associated with impaired mitosis [35]. Therefore, we used data from the Mitocheck project database (http://www.mitocheck.org/) [36,37] to determine the enrichment (or depletion) ratio of our bait sets for each mitotic phenotype scored in Mitocheck (Appendix A). We then used the enrichment ratios to cluster bait sets according to their functional similarities (Figure 3D). Bait sets clustered around two large components according to their involvement in mitotic processes. One cluster (NS1, NS2A, and NS3 bait sets) presented enrichment of mitotic phenotypes, while the second (NS2B, NS4A, NS4B, and NS5) did not, suggesting that NS1, NS2A, and NS3 are more likely to disrupt cellular mitotic processes (Figure 3D).

Finally, to identify individual preys more likely to be involved in mitotic processes, we clustered all preys according to their Mitocheck enrichment rations (Appendix A) and identified a cluster of nine proteins (CEP192, FAM184A, PAPSS1, EFTUD2, ZNF155, BAG6, SELENOP, KIF4A, and PHPT1) with phenotypes consistent with centrosomal abnormalities (Appendix A). This analysis reflected the clustering pattern for NS1, NS2A, and NS3 bait sets obtained when clustering for GO biological processes and cellular components (Figure 3B–D).

### 3.6. Integration with Other ZIKV PINs

Our work builds on three previous physical interaction networks of host and ZIKV proteins [19,20,21]. We integrated our PIN with the published networks to evaluate the level of overlap between the four PINs (Figure 4A; Appendix A). No common hit was shared by all four PINs and pair-wise overlaps ranged from 1 to 50 hits, suggesting that the ZIKV–host protein interacting network is still far from reaching saturation (Figure 4A,B).

We identified five highly internally connected clusters among the integrated PIN (Figure 4C). All five clusters contained components of the ZIKV PIN from this study. Four of them were enriched in proteins involved in the following: (a) anaphase promoting complex (APC)-dependent proteasomal ubiquitin-dependent protein catabolic process (GO31145); (b) protein amino acid N-linked glycosylation via asparagine (GO18279); (c) protein folding (GO6457); (d) regulation of transcription (GO45449); and (e) histone H2B ubiquitination (GO33523).

Finally, 14 unique nodes of our merged ZIKV PIN (9.3% of the data set) have been shown to be important for proper replication of different Flavivirus (Appendix A) [38,39,40], suggesting that our network also contains proteins that could explain the mechanisms of ZIKV replication and help identify therapeutic targets.

### 3.7. Integration with a Microcephaly-Associated Network

We generated a new network composed of microcephaly-associated genes/proteins (Appendix A) and our merged ZIKV PIN using BisoGenet to impute known interactions between nodes in this network (see Experimental Procedures) (Figure 5A). Four highly cohesive (i.e., highly connected internally, but only sparsely with the rest of the network) clusters emerged (Figure 5B). Of those, only three contained components of the ZIKV PIN: anaphase promoting complex (APC)-dependent proteasomal ubiquitin-dependent protein catabolic process (GO31145), centrosome duplication (GO7099), and COPI coating of Golgi vesicle (GO48205). Finally, we identified four nodes common to both PINs (CEP192, ASXL1, VARS, and EFTUD2). A similarly limited overlap between the microcephaly-associated and merged ZIKV PIN was also obtained with the three other previously determined ZIKV PINs (Figure 5C).

### 3.8. PIAS1 Modulates NS5 Protein Stability

The Y2H screening identified PIAS1 (protein inhibitor of activated STAT1) as an interacting partner of NS5. We confirmed this interaction using a GST-pulldown assay in HEK293FT cells co-transfected with GST-tagged PIAS1 and GFP-tagged NS5 (Figure 6A).

PIAS1 is an E3 SUMO-protein ligase implicated in the maintenance of protein stability [41,42]. Curiously, SUMOylation of the DENV NS5 has been linked to its stabilization, thus stimulating viral replication in human cells [43]. This observation prompted us to evaluate the impact of PIAS1 overexpressing on ZIKV NS5 protein stability. HEK293FT cells were transfected with GST-tagged PIAS1 and GFP- or CBP-tagged NS5 constructs and treated with cyclohexamide (CHX) to inhibit de novo protein synthesis. NS5 half-life was evaluated in a time course (Figure 6B). PIAS1 overexpressed is correlated with a decrease in NS5 protein stability, as seen 12 h after CHX treatment (compare lines 4 and 8). PIAS1 modulates steady state levels of NS5 even at lower expression levels of PIAS1 (Figure 6C) and levels of NS5 decrease in a PIAS1 dose-dependent manner (Figure 6D). Taken together, our data suggest that PIAS1 is involved in modulating the stability of ZIKV NS5.

## 4. Discussion

In humans, ZIKV infection was correlated with congenital microcephaly in newborns and with other neurological conditions in adults [4,5,6,7,10,11,12]. Still, little is known about the molecular mechanism of ZIKV infection and how it relates to neurological disorders. Here, we present a human host protein–ZIKV (Brazilian genotype) NS protein interaction network. This network was obtained by a combination of yeast two-hybrid (Y2H) screens and tandem affinity purification coupled to mass spectrometry (TAP-MS). Y2H screens primarily reveal direct pair-wise interactions and are capable of detecting transient interactions, while TAP-MS will reveal proteins engaged in stable complexes, which will eventually result in the identification of indirect interactions [44,45,46,47]. The use of both methods results in a comprehensive panorama of ZIKV protein–protein interactions.

The merged network combining two complementary methods (Y2H and TAP-MS) contains 157 nodes and 189 interactions with a limited overlap between the two methods, consistent with other previously determined PIN [23,46,47]. Further, the subset of Y2H interactions validated in human cells displayed a false positive rate of ~24% (9/38) averaged across all seven ZIKV NS proteins, as judged by confirmation interaction experiments using SBP pulldown assays. This is in line with other published Y2H screens [23,48,49,50,51]. These results suggest that this PIN contains high-confidence interactions.

Twenty-nine human proteins interacted with more than one ZIKV protein (19.3% of all hits). Similar relatively high levels of promiscuity of human proteins in relation to their viral interactors were also found in previous studies. Scaturro et al. [19] and Coyaud et al. [20] had 10.5% and 36% of all hits interacting to more than one ZIKV protein, respectively. Although Shah et al. [21] had a much lower (0.3% of all hits) rate of human proteins interacting to more than one ZIKV protein, a high level of promiscuity of human proteins is also apparent across studies, where the same human proteins are often found interacting with distinct ZIKV proteins. For example, all human proteins shared between Shah et al. [21] and Scaturro et al. [19], or between Shah et al. and this study, were found to interact with different ZIKV NS proteins. In addition, comparisons across other studies showed consistently high levels of discordance in bait interactions (Figure 4B). These data suggest that different ZIKV NS proteins have common targets in the human proteome. However, it is unclear why different studies detected exclusive interactions with different baits. It is conceivable that several ZIKV NS proteins interact with large protein complexes, such as the 26S subunit of the proteasome complex, via different targets; furthermore, differences in the biology of the cells providing the proteome (i.e., levels of protein expression and formation of specific protein complexes), the biochemical methods, or the filtering criteria for significant interactions may also determine which interactions are robust enough to result in detection.

Several aspects could account for the low level of overlapping between studies. Although every study used a different cell line, three of them used derivatives of HEK293 cells (HEK 293T, HEK 293FT, HEK 293 T-rex) and one used SK-N-BE2 neuroblastoma cells [19]. Yet, low overlap was also encountered in pair-wise comparison between studies with HEK cells (Shah et al. and Golubeva et al.; Shah et al. and Coyaud et al.). Conversely, a slightly higher overlap can be found between studies with HEK cells (Golubeva et al. and Coyaud et al.), as well as with studies with different cell lines (Coyaud et al. and Scaturro et al.). These observations suggest that differences in cell lines are unlikely to explain the low overlap.

In addition to affinity purification followed by mass spectrometry used in every study, complementary methods were also used such as BioID (a proximity-dependent labeling approach) [20] and the yeast two-hybrid (this study) that could partly explain the differences between studies. Other small differences in methods (expression vectors, affinity tags) and preys (all viral proteins versus only non-structural proteins; amino acid sequence differences between virus strains used) could also contribute to the differences across studies. Alternatively, some could represent spurious interactions detected as a result of the overexpression of the baits; however, all four studies used stringent cut-off measures and validation experiments, ensuring that the number of false positive results is likely to be low. Furthermore, the limited number of proteins in our dataset with high CRAPome [27] scores indicating consistent recovery in affinity proteomics as non-specific background also suggests that the differences across studies are unlikely to be explained by a large number of false positives. We propose that the low overlap among these studies suggests that they have not reached saturation and other ZIKV protein-interacting host proteins are still to be discovered.

Further, we identified multiple components of CCT (chaperonin containing TCP1 or TriC-TCP-1 ring complex) complex as targets. This complex plays a role in trafficking of telomerase and small Cajal body (CB) RNAs through the proper folding of the telomerase cofactor, TCAB1 [52]. CBs are transcription-dependent nuclear compartments and play a critical role in neuron biology through snRNP and snoRNP assembly [53]. Interestingly, Coyaud et al. [20] demonstrated that ZIKV NS5 expression leads to an increase in the absolute number of CBs per cell, but to a reduction of the volume of these CBs, suggesting that NS5 expression could lead to CB fragmentation. Our data point to the interaction of NS1 with multiple components of the CCT complex, suggesting that NS1 could also play a role in CB stability and in neural disorders. Additionally, it has already been shown that the Dengue virus (DENV) infection occurs in an NS1/CCT-dependent manner [54].

Centrosomal abnormalities lead to impaired mitosis, which is a hallmark of MCPH. In fact, our data set presents multiple proteins related to phenotypes associated with impaired mitosis (Figure 3B,C). Furthermore, our PIN shares 24 (16% of all unique hits) known interaction partners of 14 (out of 18) MCPH loci plus CEP63 (Appendix A).

In that context, CEP192 (identified as an NS3 interaction partner by Y2H) plays a central role in the initial steps of centriole duplications through the interaction and recruitment of CEP152 (MCPH9) and PLK4, respectively, which is necessary for the proper recruitment of SAS6 (MCPH14), STIL (MCPH7), and CENPJ (MCPH6) [55,56,57,58,59,60]. Our data suggest that NS3 could interfere with centriole duplication and, consequently, could be important for the ZIKV-associated microcephaly phenotype. Furthermore, GO enrichment analysis and Mitocheck phenoclusters suggest that NS1, NS2A, and NS3 target host factors are implicated in mitotic phenotypes.

In humans, viral infection activates the type-I interferon (IFN-I) signaling leading to STAT1/2 activation. The ZIKV5 protein acts as an antagonist of the IFN-I pathway by stimulating STAT2 (but not STAT1) degradation [61]. STAT1 activity is modulated by PIAS1, which has been implicated in herpes simplex viral replication [62]. We identified PIAS1 as an interacting partner of NS5 and showed that overexpression of PIAS1 results in a shorter NS5 protein half-life. Our data suggest that PIAS1 can modulate the levels of ZIKV NS5, but it is unclear the extent to which this modulation may affect ZIKV replication. Interestingly, a CRISPR/Cas9 screening revealed that PIAS1-depleted cells are more sensitive to ZIKV infection-dependent lethality [38]. Collectively, these data suggest that PIAS1 might play an important role in ZIKV biology by modulating NS5 protein levels.

In summary, the data presented here together with three previously published studies [19,20,21] provide a valuable resource to dissect the mechanistic underpinnings of central nervous system perturbations caused by ZIKV infection and to identify potential pharmacological targets. A small number of overlapping hits across different studies suggest that the screens are still far from reaching saturation.

## Figures and Tables

**Figure 1 cells-09-00153-f001:**
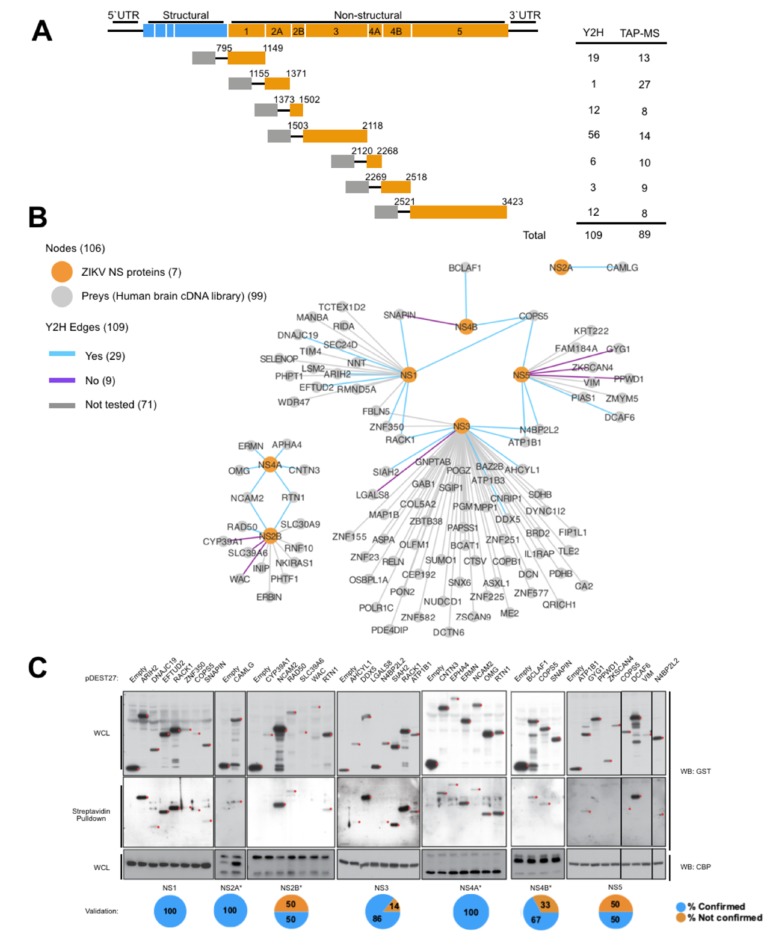
Yeast two-hybrid (Y2H) protein–protein interaction network (PIN). (**A**) Schematic representation of the ZIKA virus (ZIKV) genome and non-structural (NS) constructs used on our protein–protein interaction screenings. Grey boxes represent the GAL4 DNA binding domain (DBD) and the streptavidin binding protein (SBP)–calmodulin binding protein (CBP) tag used on the yeast two-hybrid (Y2H) and tandem affinity purification (TAP) assays, respectively. The number of hits identified by each assay and bait is summarized on the right. UTR, untranslated region. (**B**) Network of the interactions identified by Y2H screens. ZIKV NS nodes are colored in orange, and gray nodes indicate other proteins that were recovered as preys. Edge colors represent results from validation. Edges to all preys are shown and ‘not validated’ (No), ‘validated’ (Yes), and ‘not tested interactions’ (Not tested), and are denoted by blue, purple, and gray edges, respectively. The color legend is depicted on the upper left-hand corner. (**C**) Streptavidin pulldown of TAP-tagged NS constructs from 293FT whole cell lysates followed by Western blotting with the indicated antibodies. The individual percentage of hits validated by bait is depicted on the right. Red dots indicate the expected band size. Red asterisk for Streptavidin pulldown assay.

**Figure 2 cells-09-00153-f002:**
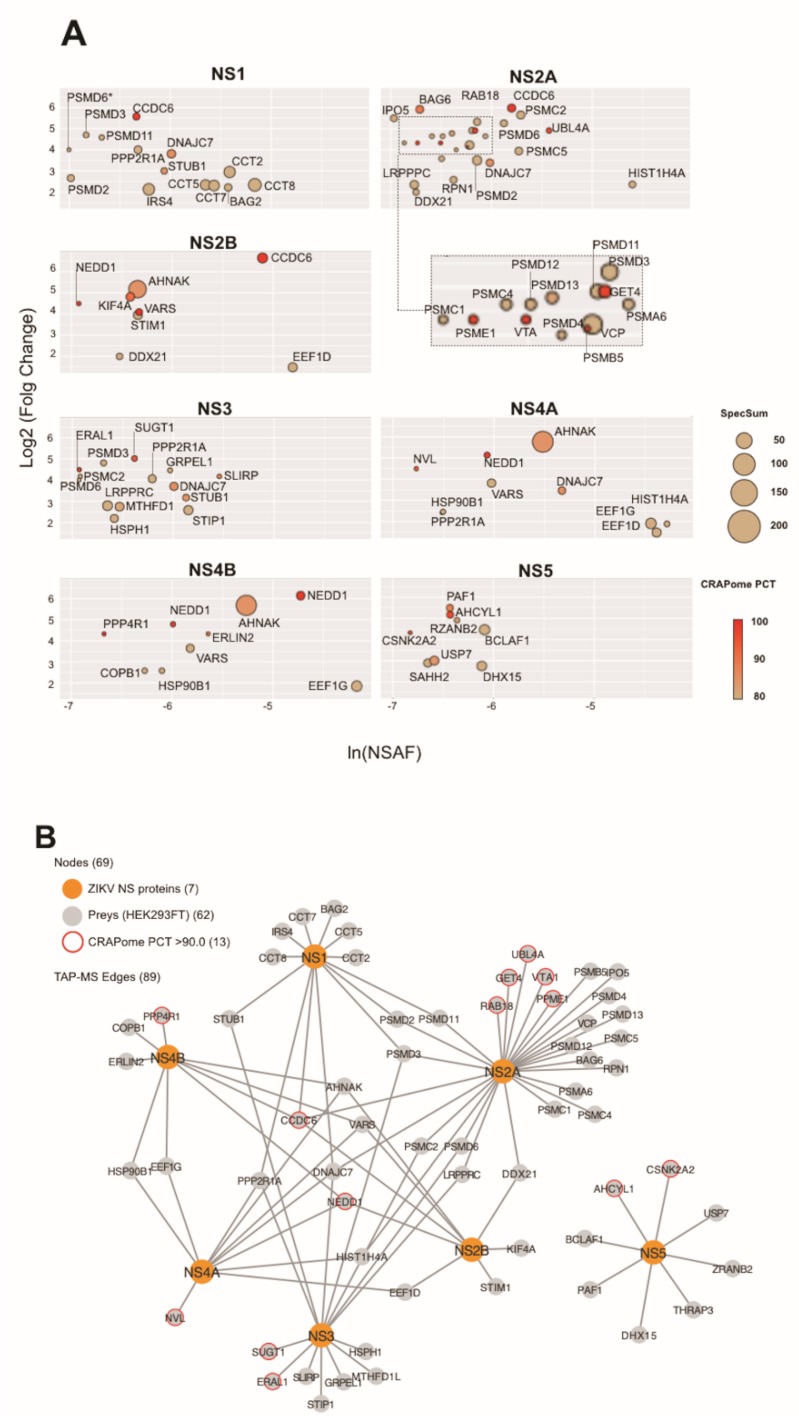
TAP-MS PIN. (**A**) Protein interaction profile of TAP-MS screenings plotted based on normalized spectral abundance factor (NSAF) [34] (X axis) and specificity based on fold change of spectral counts (Y axis) between TAP-tagged NS proteins and TAP-tagged GFP (negative control). Node size denotes the spectral sum (Spec sum) obtained for each protein. Node color denotes CRAPome PCT score according to the scale. (**B**) Network of the interactions identified by TAP-MS screens. Orange nodes represent ZIKV NS proteins (baits) and gray nodes represent human proteins (preys). Gray nodes with a red circle indicate a prey with high CRAPome score.

**Figure 3 cells-09-00153-f003:**
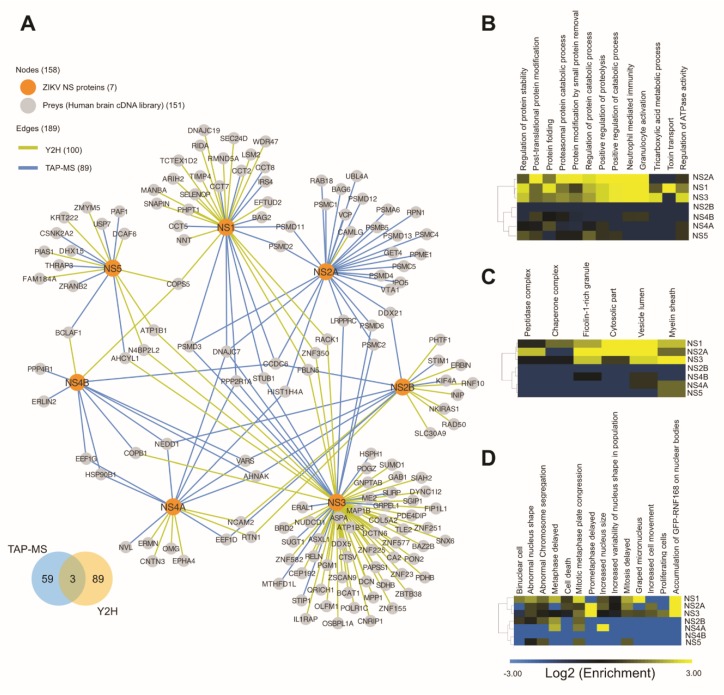
Merged ZIKV PIN. (**A**) Network of the interactions identified by Y2H and TAP-MS screens. The color legend is depicted on the upper left-hand corner. (**B**,**C**) Clustering of bait sets according to gene ontology (GO) enrichment ratio for biological processes (**B**) and cellular component (**C**). (**D**) Phenoclusters (clustering of bait sets according to enrichment or depletion of Mitocheck phenotype classes. Clustering and visualization were performed using Cluster v3.0 software and TreeView v1.1.6r4, respectively.

**Figure 4 cells-09-00153-f004:**
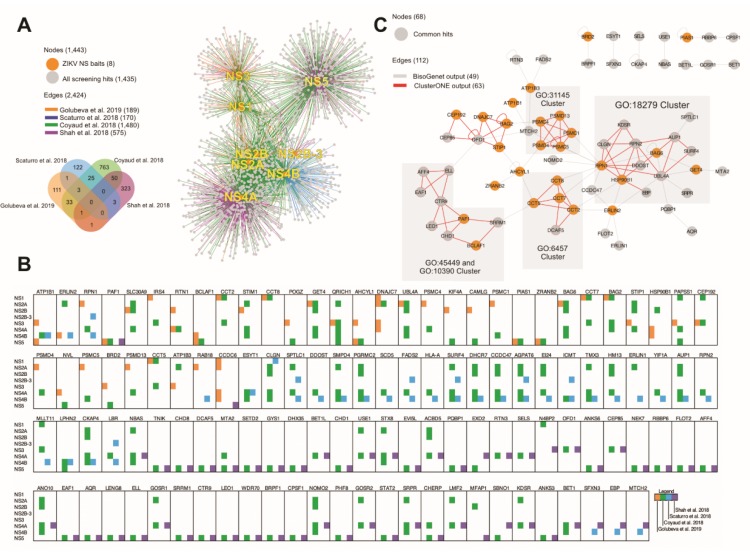
Integration of ZIKV PINs. (**A**) Network of the interactions identified by this study, Scaturro et al. 2018, Shah et al. 2018, and Coyaud et al. 2018. The color legend is depicted on the upper left-hand corner. (**B**) Graphic representation of common preys according to the bait and study in which they were identified. (**C**) Clustering of bait sets according to overlapping protein complexes among the integrated network using ClusterONE (Version 1.0) [29]. Gene ontologies of these networks were obtained by the Cytoscape plugin, BINGO [30]. GO accession numbers represent the following biological process: GO31145—anaphase promoting complex (APC)-dependent proteasomal ubiquitin-dependent protein catabolic process, GO18279—protein amino acid N-linked glycosylation via asparagine, GO6457—protein folding, GO45449—regulation of transcription, and GO33523—histone H2B ubiquitination. Proteins identified in this study are represented as orange nodes.

**Figure 5 cells-09-00153-f005:**
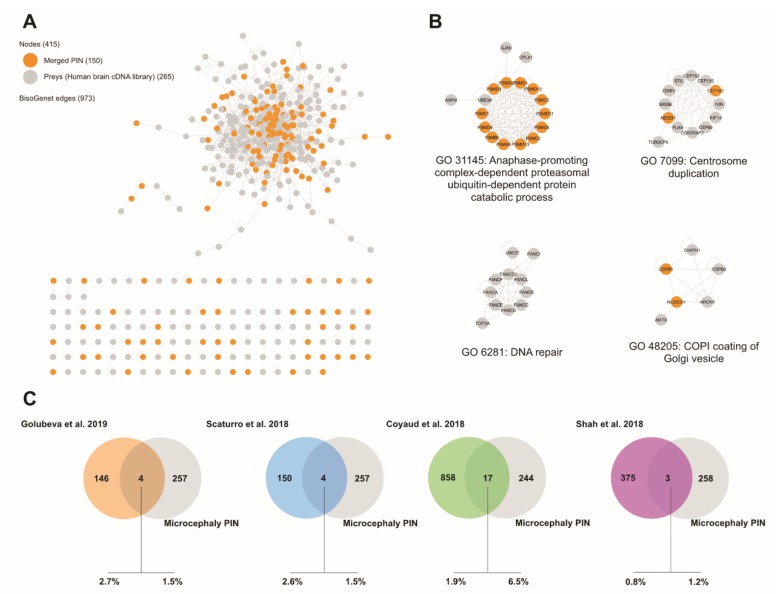
Integration of Microcephaly-associated PIN with the merged ZIKV PIN. (**A**) Network of the interactions identified by this study, and proteins related to the Microcephalic phenotype (see Experimental Procedures). (**B**) Clustering of bait sets according to overlapping protein complexes among the integrated network using ClusterONE (Version 1.0) [29]. Gene ontology of these networks were obtained by the Cytoscape plugin, BINGO [30]. Proteins identified in this study are represented as orange nodes. (**C**) Venn diagrams represent the overlap between the Microcephaly-associated PIN and the individual ZIKV PINs.

**Figure 6 cells-09-00153-f006:**
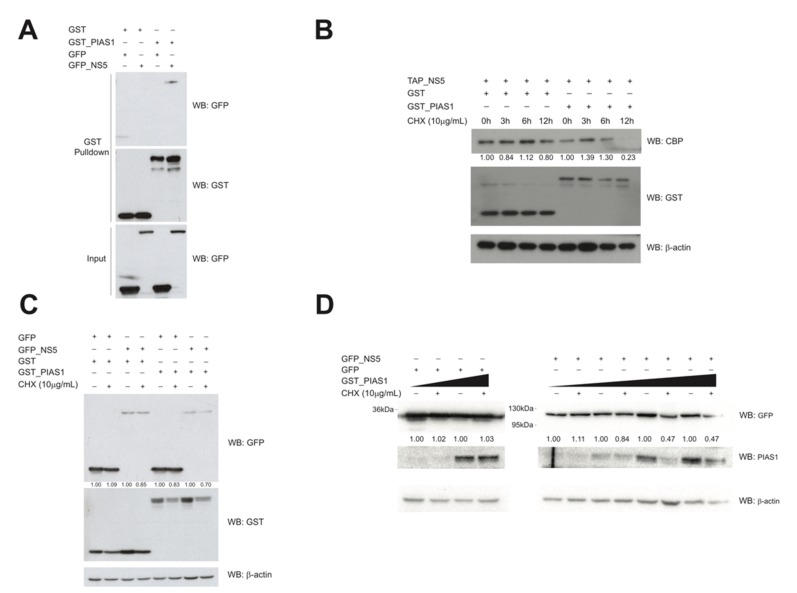
PIAS1 interacts with and modulates ZIKV NS5 protein stability. (**A**) GST pulldown was conducted using whole cell lysates of HEK293FT cells transfected with GST-tagged PIAS1 and GFP-tagged NS5 (empty vectors were used as negative controls). Input represents 10% of the lysate used in the GST pulldown assay. (**B**) HEK293FT cells transfected with GST-tagged constructs (empty vector or PIAS1) and pNTAP NS5 and treated with 10 µg/mL cyclohexamide (CHX) for the indicated time points. (**C**) HEK203FT cells transfected with GST and GFP-tagged constructs. Cells were treated, 24 h after transfections, with 10 µg/mL cyclohexamide (CHX) for 12 h. (**D**) HEK293FT cells were transfected with GFP-tagged constructs (empty vector or NS5) and different amounts of GST-tagged PIAS1 cDNA (0.5 µg, 1 µg, 2 µg, or 4 µg). pQCXIH was used to normalize the amount of transfected DNA. At 20 h post-transfection, cells were treated with 10 µg/mL cyclohexamide (CHX) for 12 h. + and – signs indicate presence or absence of the reagent indicated on the left, respectively.

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
