# Peer review of "Network of Interactions between ZIKA Virus Non-Structural Proteins and Human Host Proteins"

_cells, 2020, doi:10.3390/cells9010153_

Round 1

Reviewer 1 Report

In the current work, Golubeva et al present a large scale data set detailing the interactome of Zika virus non-structural proteins with human cells. In general, the manuscript is extremely well written and data well presented. As is currently noted by the authors, the work does duplicate 3 previously published papers, however, given the large scale nature of the experiments, this is not seen as something which should preclude publication, as such I have only minor comments.  

In figure 1C the pull down blots shown do not support the conclusions by the authors as to the percentage of validated hits. For example, the NS5 blot analyzes 8 potential interactions and says it validates 4 (50%) but only 1 band is obviously seen on the blot. I’m guessing this is due to exposures, but the authors should present data which better supports their conclusions.

Some additional discussion as to why this works results differ so much from the previous 3 works would be beneficial. Were different strains used? Different cells? Or is it just due to technical variation?

Author Response

In the current work, Golubeva et al present a large scale data set detailing the interactome of Zika virus non-structural proteins with human cells. In general, the manuscript is extremely well written and data well presented. As is currently noted by the authors, the work does duplicate 3 previously published papers, however, given the large scale nature of the experiments, this is not seen as something which should preclude publication, as such I have only minor comments.

Response: We appreciate that the reviewer recognizes the high quality of the
manuscript. We agree that three previous papers have reported similar approaches, yet there is little overlap among those four studies. The lack of overlap is evident also in pair-wise comparison of all four studies, suggesting that instead of having some studies with low quality, the lack of overlap may be due to differences in experimental conditions and the fact that none of these studies has reached saturation. We now discuss in more detail the experimental differences between studies and argue that all four studies can contribute to our knowledge of Zika virus. In addition to an additional set of proteins identified beyond the other studies, the following differences set our study apart:
 - Inclusion of data from yeast-two hybrid, a complementary method to identify
(mostly) direct pair-wise interactions
 - Use of the Brazilian sequence to conduct the experiments
 - Provides preliminary evidence on the role of PIAS1 in the regulation of NS5
 - Phenocluster analysis
 - Integration with Microcephaly PIN

In figure 1C the pull down blots shown do not support the conclusions by the authors as to the percentage of validated hits. For example, the NS5 blot analyzes 8 potential interactions and says it validates 4 (50%) but only 1 band is obviously seen on the blot.
I’m guessing this is due to exposures, but the authors should present data which better supports their conclusions.

Response: We have adjusted the contrast in the figure to make the weaker bands more apparent. This is now shown on revised Figure 1C.

Some additional discussion as to why this works results differ so much from the
previous 3 works would be beneficial. Were different strains used? Different cells? Or is it just due to technical variation?

Response: We thank the reviewer for the comment, and appreciate the opportunity to expand our discussion on the possible reasons for the lack of overlap. We have expanded the discussion about the differences across the four studies (added lines 430-450; line numbering refers to the clean revised version). See also response to item (1) from Reviewer #2:

Response 1 to Referee 2: We agree with the reviewer that this point merits further discussion. We would like to point out that the lack of overlap is evident in any pair-wise comparison between studies. As all studies have used relatively stringent cut-offs for filtering and for deciding which interactions are significant we believe this is unlikely the result of false positives. If the lack of overlap had been found only between ours and other studies, but not in pair-wise comparisons between the other studies, then there would be an indication that our study suffered from high numbers of false positive. We propose that most differences are due to the fact that none of the studies have reached saturation. We have extended our discussion to include assessments of host cell, methods, and analytical thresholds (lines 430-450; line numbering refers to the clean revised version).

Reviewer 2 Report

The study provides interesting data on protein interactions between Zikavirus NS proteins and human proteins.

The overlap between the current study and the other studies cited by the authors is extremely limited. Although mentioned in the discussion, this point needs to be discussed in more detail. How high is the probability that the limited overlap is due to false positives?

Line 90: site-directed mutagenesis is mentioned. The rationale for this should be explained

Lines 115, 136 : should be 10 exp6 and 10 exp8, respectively

Line124: SBP and CBP: abbreviations should be explained

Line 184-185“Enrichment ratios that had a 0.0 value were replaced by 0.3 (half of the

lowest non-zero value, 0.6) in the complete set.” The rationale for this should be explained.

Figure 1: Figure 1B should contain only validated binding partners. Please explain in the legend what is meant by edges “yes” , “no”. 1C: please explain “WCL”.Red dots are mentioned but cannot be found.

Figure 2 A Please explain legend.

Line 354 and following: false positive rate of 24%. Please indicate details.

Author Response

Reviewer #2
The study provides interesting data on protein interactions between Zikavirus NS
proteins and human proteins. The overlap between the current study and the other studies cited by the authors is extremely limited. Although mentioned in the discussion, this point needs to be discussed in more detail. How high is the probability that the limited overlap is due to false positives?

Response: We agree with the reviewer that this point merits further discussion. We would like to point out that the lack of overlap is evident in any pair-wise comparison between studies. As all studies have used relatively stringent cut-offs for filtering and for deciding which interactions are significant we believe this is unlikely the result of false positives. If the lack of overlap had been found only between ours and other studies, but not in pair-wise comparisons between the other studies, then there would be an indication that our study suffered from high numbers of false positive. We propose that most differences are due to the fact that none of the studies have reached saturation. We have extended our discussion to include assessments of host cell, methods, and analytical thresholds (lines 430-450; line numbering refers to the clean revised version).

Line 90: site-directed mutagenesis is mentioned. The rationale for this should be
explained

Response: We failed to obtain the correct product corresponding to NS3 and NS5 using the Brazilian virus isolate as template for PCR. Therefore, to obtain the Brazilian genotype we introduced nucleotide substitutions by site-directed mutagenesis using the Asian genotype NS3 and NS5 cDNAs as a starting point. This is now described in the Materials and Methods section (lines 91-95).

Lines 115, 136 : should be 10 exp6 and 10 exp8, respectively

Response: We thank the reviewer for catching this. Although it shows correctly in the word document it reverts to an incorrect format upon pdf conversion. We have further raised the exponent by 3pt. It seems to have solved the issue.

Line124: SBP and CBP: abbreviations should be explained

Response: The abbreviations SBP (Streptavidin-binding peptide) and CBP
(Calmodulin-binding peptide) are now explained (line 98).

Line 184-185“Enrichment ratios that had a 0.0 value were replaced by 0.3 (half of the lowest non-zero value, 0.6) in the complete set.” The rationale for this should be explained.

Response: In order to depict increases and decreases as numerically equivalent
enrichment rations were log2 transformed. To allow for log transformation, 0 values are commonly replaced (undefined log2) by non-zero values, using a variety of techniques: such as replacing by log(x+1) or by log(X - (min(X) - 1)). We used another commonly used technique which uses the smallest value of the dataset and divides by 2. We have revised the explanation in the Material and Methods in the following way (lines 191-193):

“Enrichment ratios were log2-transformed to depict increase and decrease changes as numerically equal but with opposite sign. To allow for log transformation Enrichment ratios with a 0.0 value were replaced half of the lowest non-zero value in the complete set. Bait sets were clustered with Cluster 3.0 using Correlation (uncentered) metric of similarity with no filtering. The
clustering method chosen was complete linkage. It was visualized using Java TreeView v1.1.6r4 [32].”

Figure 1: Figure 1B should contain only validated binding partners. Please explain in the legend what is meant by edges “yes” , “no”. 1C: please explain “WCL”.Red dots are mentioned but cannot be found.

Response: We argue that there is a rationale for showing all partners, not only the validated ones. This allows for the view of the complete set of results, even the ones that were not assessed for validation. Also, because conditions were not optimized for the validation of every interaction some interactions that were not validated could be bona-fide interactions. We stress that these interactions that were not validated were not part of the downstream analysis. We suggest that the full set of interactors should be shown in which the edges show their status. Another approach, if the reviewer feels strongly about only showing the validated hits in Figure 1, would be to show all interactions (annotated according to their validation status) in a Supplementary Figure.
Please let us know if the alternative solution is preferable and we will provide the revised figures.
The revised legend now explains ‘yes’ and ‘no’ edges and WCL. The red dots can now be seen in the revised figure. We tried to keep the dots small not to interfere with the image of the bands, but we can increase their size according to the editorial guidelines.

Figure 2 A Please explain legend.

Response: The legend to Figure 2 is now explained (lines 298-304):
Figure 2. TAP-MS PIN. A. Protein interaction profile of TAP-MS screenings plotted based on Normalized Spectral Abundance Factor (NSAF) [37] (X axis) and specificity based on fold change of spectral counts (Y axis) between TAP-tagged NS proteins and TAP-tagged GFP (negative control). Node size denotes the spectral sum (Spec sum) obtained for each protein. Node color denotes CRAPome PCT score according to the scale. B. Network of the interactions
identified by TAP-MS screens. Orange nodes represent ZIKV NS proteins (baits) and gray nodes represent human proteins (preys). Gray nodes with a red circle indicate a prey with high CRAPome score.

Line 354 and following: false positive rate of 24%. Please indicate details.

Response: We have revised the sentence to indicate details as follows (lines 408-410):“Further, the subset of Y2H interactions validated in human cells displayed a false positive rate of ~24% (9/38) averaged across all seven ZIKV NS proteins as judged by confirmation interaction experiments using SBP pull down assays. This is in line with other published Y2H screens [23,48-51].”

Reviewer 3 Report

In this study, Golubeva et al investigate the interaction of the Zika virus non-structural proteins with the proteins of the human host. They identify the protein-protein interaction network via two different approaches. First they perform a yeast-two-hybrid screen with the tagged ZIKV NS proteins and a human neuronal cDNA prey library. Second they pull down tagged ZIKV NS proteins and ID bound host proteins via mass-spec. The authors then incorporate these findings into protein-interaction networks. Overall the authors identify 189 hits from the two different techniques with only three hits overlapping with the two techniques.

This study is not novel as three other manuscripts have been published which map Zika-host PIN. The lack of novelty is not a concern as data replication is key to scientific progress; however, there is little/no overlap of the findings of this study with that of the published manuscripts. These discrepancies highlight the need to validate PIN hits in the context of a viral infection. The authors could either perform a targeted Co-IP experiment with antibodies specific for a Zika viral protein from infected cell lysate. Alternatively the authors could knock-down hits from the PIN and demonstrate an effect on viral replication. Without any efforts to validate hits resulting from these screens, I cannot recommend this study for publication.

Other minor concerns/corrections are detailed below:

Line 65: African lineage Zika virus has also been shown to be damaging to fetus in small (PMID: 30995223) and large animal models (PMID: 31340725) Line 45: Aedes aegypti is the dominant vector, although the virus can be carried by albopictus. Line 113: The human brain cDNA prey library from ClonTech is catalog #: 630486 Line 213: Figure 1 shows SBP pulldown while text here states CBP pulldown Figure 4: B and C are switched according to the legend

Author Response

Reviewer #3

In this study, Golubeva et al investigate the interaction of the Zika virus non-structural proteins with the proteins of the human host. They identify the protein-protein interaction network via two different approaches. First they perform a yeast-two-hybrid screen with the tagged ZIKV NS proteins and a human neuronal cDNA prey library. Second they pull down tagged ZIKV NS proteins and ID bound host proteins via massspec. The authors then incorporate these findings into protein-interaction networks. Overall the authors identify 189 hits from the two different techniques with only three hits overlapping with the two techniques.

This study is not novel as three other manuscripts have been published which map
Zika-host PIN. The lack of novelty is not a concern as data replication is key to
scientific progress; however, there is little/no overlap of the findings of this study with that of the published manuscripts. These discrepancies highlight the need to validate PIN hits in the context of a viral infection. The authors could either perform a targeted Co-IP experiment with antibodies specific for a Zika viral protein from infected cell lysate. Alternatively the authors could knock-down hits from the PIN and demonstrate an effect on viral replication. Without any efforts to validate hits resulting from these screens, I cannot recommend this study for publication.

Response: We agree that the general approach (i.e. to generate a map of ZIKV-host protein interactions. However, all studies were conducted independently and report sets of proteins with a low overlap. We argue that these differences suggest the screens have not yet reached saturation. Please see response to item (1) from Reviewer #1 for other aspects in which our study differs from the previous studies.

While we do agree that further validation and exploration of these interactions are necessary in a more physiological setting, our laboratory is not equipped or certified to conduct live virus research. Our paper aims to provide a roadmap for further exploration but believe that such important validation step is beyond the scope of the present manuscript.

We now also present a vignette that validates functionally the interaction with PIAS1 and suggests a role in the stability of NS5. We believe this interaction would constitute a clear proof-of-principle for further validations.

Line 65: African lineage Zika virus has also been shown to be damaging to fetus in small (PMID: 30995223) and large animal models (PMID: 31340725).

Response: We apologize for the oversight. The revised text is corrected and the
suggested references were incorporated (line 66).

Line 45: Aedes aegypti is the dominant vector, although the virus can be carried by albopictus.

Response: The revised text now mentions only the dominant vector (line 46).

Line 113: The human brain cDNA prey library from ClonTech is catalog #: 630486.

Response: We thank the reviewer for catching this. It is now corrected in the revised text (line 121).

Line 213: Figure 1 shows SBP pulldown while text here states CBP pulldown Figure 4: B and C are switched according to the legend.

Response: We apologize for the oversight. This has now been corrected (line 236).